

# TROPESS/CrIS carbon monoxide profile validation with NOAA GML and ATom in situ aircraft observations

Helen M. Worden[1], Gene L. Francis[1], Susan S. Kulawik[2], Kevin W. Bowman[3], Karen Cady-Pereira[4], Dejian Fu[3], Jennifer D. Hegarty[4], Valentin Kantchev[3], Ming Luo[3], Vivienne H. Payne[3], John R. Worden[3], Róisín Commane[5], Kathryn McKain[6,7]

[1]Atmospheric Chemistry Observations and Modeling (ACOM), National Center for Atmospheric Research (NCAR), Boulder, CO, USA
[2]BAER Institute, 625 2nd Street, Suite 209, Petaluma, CA, USA
[3]Jet Propulsion Laboratory / California Institute for Technology, Pasadena, CA, USA
[4]Atmospheric and Environmental Research Inc., Lexington, MA, USA
[5]Dept. of Earth and Environmental Sciences, Lamont-Doherty Earth Observatory, Columbia University, Palisades, NY, USA
[6]Cooperative Institute for Research in Environmental Sciences (CIRES), University of Colorado, Boulder, CO, USA
[7]Global Monitoring Division (GMD), National Oceanic and Atmospheric Administration, Boulder, CO, USA

*Correspondence to*: Helen Worden (hmw@ucar.edu)

**Abstract.** The new single pixel TROPESS (TRopospheric Ozone and its Precursors from Earth System Sounding) profile retrievals of carbon monoxide (CO) from the Cross-track Infrared Sounder (CrIS) are evaluated using vertical profiles of in situ observations from the National Oceanic and Atmospheric Administration (NOAA) Global Monitoring Laboratory (GML) aircraft program and from the Atmospheric Tomography Mission (ATom) campaigns. The TROPESS optimal estimation retrievals are produced using the MUSES (MUlti-SpEctra, MUlti-SpEcies, MUlti-Sensors) algorithm which has heritage from retrieval algorithms developed for the EOS/Aura Tropospheric Emission Spectrometer (TES). TROPESS products provide retrieval diagnostics and error covariance matrices that propagate instrument noise as well as the uncertainties from sequential retrievals of parameters such as temperature and water vapor that are required to estimate the carbon monoxide profiles. The validation approach used here evaluates biases in column and profile values and the validity of the retrieval error estimates using the mean and variance of the compared satellite and aircraft observations. CrIS-NOAA GML comparisons had biases of 0.6 % for partial column average volume mixing ratios (VMR) and (2.3, 0.9, -4.5) % for VMR at (750, 511, 287) hPa vertical levels, respectively, with standard deviations from 9 % to 14 %. CrIS-ATom comparisons had biases of -0.04 % for partial column and (2.2, 0.5,-3.0) % for (750, 511, 287) hPa vertical levels, respectively, with standard deviations from 6 % to 10 %. The reported observational errors for TROPESS CrIS CO profiles have the expected behavior with respect to the vertical pattern in standard deviation of the comparisons. These comparison results give us confidence in the use of TROPESS CrIS CO profiles and error characterization for continuing the multi decadal record of satellite CO observations.



## 1. Introduction

Carbon monoxide (CO) is a useful tracer of atmospheric pollution with direct emissions from incomplete combustion such as biomass and fossil fuel burning and secondary production from the oxidation of methane (CH4) and volatile organic compounds (VOC). Atmospheric CO distributions have a seasonal cycle that is mainly driven by photochemical destruction, which allows CO to build up over winter and early spring in higher latitudes. The lifetime of CO, weeks to months, (e.g., Holloway et al., 2000), is long enough to allow observations of pollution plumes and their subsequent long range transport, but short enough to distinguish the plumes against background seasonal distributions (e.g., Edwards et al., 2004, 2006; Hegarty et al., 2009, 2010). As a dominant sink for the hydroxyl radical (OH), CO plays a critical role in atmospheric reactivity (e.g., Lelieveld et al., 2016) and is considered a short-lived climate pollutant (SLCP) because of its impacts to methane lifetime and carbon dioxide and ozone formation (e.g., Myhre et al., 2014; Gaubert et al., 2017).

Global observations of tropospheric CO from satellites started in 2000 with the NASA Earth Observing System (EOS) Measurement of Pollution in the Troposphere (MOPITT) instrument on Terra (Drummond et al., 2010), followed by the EOS Atmospheric Infrared Spectrometer (AIRS, McMillan et al., 2005) on Aqua launched in 2002, the Scanning Imaging Absorption Spectrometer for Atmospheric Chartography (SCIAMACHY, de Laat et al., 2006) on Envisat launched in 2002, the EOS Tropospheric Emission Spectrometer (TES, Beer et al., 2006) on Aura launched in 2004, the Infrared Atmospheric Sounding Interferometer (IASI, Clerbaux et al., 2009) on the MetOp series beginning in 2006, the Cross-track Infrared Sounder (CrIS, Gambacorta et al., 2014) on the Suomi National Polar-orbiting Partnership (SNPP) satellite launched in 2011, and most recently the Joint Polar Satellite System (JPSS) series, TROPOMI on the Sentinel-5 precursor in 2017, (Borsdorff, et al., 2018) and the Fourier Transform Spectrometer (FTS-2) on the Greenhouse gases Observing SATellite-2 (GOSAT-2, Suto et al., 2021), launched in 2018. Satellite CO observations are assimilated for reanalyses and operational air quality forecasting (e.g., Gaubert, 2016; Inness et al., 2019; Miyazaki et al., 2020) and have been used in inverse modelling analyses to estimate emissions and attribute sources for co-emitted species such as CO2 (e.g., Kopacz et al., 2010; Jiang et al 2017; Liu et al., 2017; Zheng et al., 2019; Gaubert et al., 2020; Byrne et al., 2021; Qu et al., 2022). Trend analyses of satellite CO observations (e.g. Worden et al., 2013; Buchholz et al., 2021) show a general decline of atmospheric CO over the satellite record globally and in most regions, but with a slowing of this decrease in recent years that emphasizes the need for continued satellite CO observations that are validated and have reliable error characterization.

Similar to the recent validation study for the MUSES single pixel CO retrievals from the Aura Atmospheric Infrared Sounder (AIRS) of Hegarty et al., (2022), here we evaluate the biases and reported uncertainties of the TROPESS/MUSES CO retrievals (Bowman et al., 2021) from the Cross-track Infrared Sounder (CrIS) onboard the SNPP satellite launched in October, 2011. CrIS is a Fourier Transform Spectrometer (FTS) that has continuation instruments on the current and planned JPSS series with JPSS1/NOAA-20 launched in 2017 and planned launches in 2022, 2028 and 2032 (jpss.noaa.gov). The TROPESS CrIS CO products evaluated here use the MUSES (MUlti-SpEctra, MUlti-SpEcies, MUlti-Sensors) algorithm (Fu et al., 2016, 2018, 2019) with single field of view (FOV) radiances in sequential optimal estimation (Rodgers, 2000) retrievals of temperature, water vapor, effective cloud parameters, other trace gases and CO.



TROPESS CrIS CO products differ from other available CrIS CO products that combine 9 FOVs
to obtain a single cloud-cleared radiance and corresponding retrieval of atmospheric parameters
such as the NOAA Unique Combined Atmospheric Processing System (NUCAPS) (Gambacorta
et al., 2014, 2017) and the Community Long-term Infrared Microwave Combined Atmospheric
Product System (CLIMCAPS) (Smith and Barnet, 2020).
The MUSES algorithm was developed with heritage from Aura/TES retrieval processing and
allows for full characterization of the vertical retrieval sensitivity with an averaging kernel and
error covariance (Bowman et al., 2006). The TROPOESS/MUSES data products report a
separate matrix for the observational error terms along with the total retrieval error covariance
that includes the contribution of smoothing error. This is important for evaluation of retrieval
errors using in situ profiles since the comparison removes the effect of smoothing in the retrieval
by applying the retrieval averaging kernel and a priori to the in situ profile before differencing
(Rodgers and Connor, 2003). The TROPESS retrievals and CO data products are described in
more detail in Section 2 and the validation in situ data from the National Oceanic and
Atmospheric Administration (NOAA) Global Monitoring Laboratory (GML) aircraft network
and the Atmospheric Tomography Mission (ATom) campaigns are described in Section 3. The
validation methods are presented in Section 4 and results are shown in Section 5 with a summary
and conclusions in Section 6.
**2.  TROPESS CrIS single field of view CO profile retrievals**
The first Cross-track Infrared Sounder (CrIS) was launched 28 October, 2011 on the SNPP
satellite into a sun-synchronous polar orbit with an altitude near 830 km, and an equator-crossing
time (ascending node) near 13:30 LT. CrIS is a Fourier Transform Spectrometer (FTS) operating
in three spectral bands between 648 cm$^{-1}$ and 2555 cm$^{-1}$. This includes the R-branch of the
thermal infrared (TIR) CO (0-1) fundamental band above 2155 cm$^{-1}$. After launch, spectral
radiance data that included the CO band were collected using a spectral resolution of 2.5 cm$^{-1}$.
This resolution was relatively coarse and significantly limited the vertical sensitivity of CO
retrievals (Gambacorta et al., 2014). Following the decision to collect data at full-spectral
resolution ($\delta = 0.625$ cm$^{-1}$), these finer resolution spectral radiances have been available since 4
December 2014. Here we only consider the full-spectral resolution CrIS data.
**2.1 TROPESS retrieval approach**
TROPESS data processing (Bowman et al., 2021) produces retrievals of temperature, water
vapor and trace gases such as ozone ($O_3$), methane ($CH_4$), carbon monoxide (CO), ammonia
($NH_3$) and peroxyacetyl nitrate (PAN) from single and multiple instruments including AIRS and
OMI, CrIS and TROPOMI. Here we consider the SNPP/CrIS-only TIR CO retrievals that use the
2181-2200 cm$^{-1}$ spectral range. Bowman et al, (2021) describe the sequential MUSES retrievals
of temperature, water vaper and effective cloud properties for each FOV that are necessary for
the retrieval of CO. Each step in the sequence includes an iterative retrieval with a forward
model and updated estimate of the state vector of atmospheric parameters following the
*maximum a posteriori* (MAP) method. The forward model for radiative transfer at CrIS TIR
wavelengths uses Optimal Spectral Sampling (OSS, Moncet et al., 2015), which includes
effective cloud optical depth and height parameters (Eldering et al., 2008; Kulawik et al., 2006).
A priori profiles for TROPESS CO retrievals are taken from the model climatology used in
Aura/TES processing (MOZART, Brasseur et al., 1998), with monthly variation over a 30°



latitude and 60° longitude grid. The a priori uncertainty covariance matrix used to constrain the
retrieval is the same as used for MOPITT profiles (Deeter et al., 2010) with 30 % uncertainty for
vertical CO parameters at all levels and correlation lengths corresponding to 100 hPa between
them in the troposphere.
**2.2 TROPESS CrIS CO data examples**
Figure 1 shows an example of TROPESS/CrIS CO data for 12 September 2020 when there were
significant fires in the western US. These retrievals are from a special data collection that
processed scenes selected from 0.25°x0.25° latitude/longitude sub-sampling to enable throughput
with the available computing capacity (Bowman et al., 2021). The data in this collection are pre-
filtered for quality and Fig. 1a shows all available day and night retrievals. Fig. 1b shows the
data after higher cloudy scenes are removed (i.e, cloud tops with pressure < 700 hPa and cloud
effective optical depth > 0.1).  For reference, Fig. 1c shows the mid-tropospheric average CO
volume mixing ratio (VMR) for the a priori profiles used in the retrievals and Fig. 1d shows a
NASA Worldview (worldview.earthdata.nasa.gov) image from SNPP/VIIRS (Visible Infrared
Imaging Radiometer Suite) with clouds and smoke shown in true color and red areas indicating
fire and thermal anomalies. Since vertical profile retrievals using TIR radiances have sensitivity
to CO mainly in the free troposphere, Fig. 1 shows individual retrievals with average VMR from
vertical layers between 700 to 350 hPa. When all scenes are included, the average degrees of
freedom for signal (DFS) is 0.99 for the CrIS CO observations in Fig. 1a, and when cloudy
scenes are removed, the average DFS is 1.14 for the remaining CrIS observations in Fig. 1b.

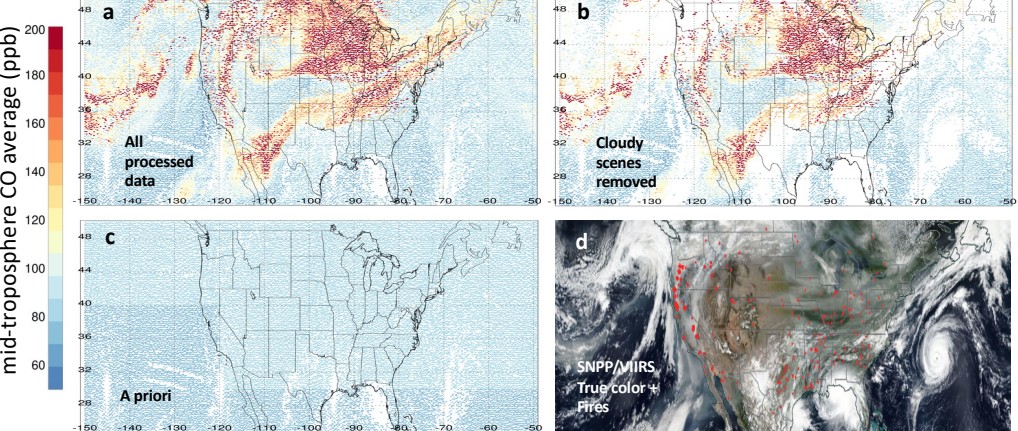

***Figure 1**. TROPESS SNPP/CrIS and SNPP/VIIRS observations for 14 September, 2020. Panel*
*(a) shows the average CO VMR for 700 to 350 hPa for all processed TROPESS CO retrievals*
*with good data quality (see text). Panel (b) shows the same free troposphere CO averages as (a)*
*but with cloudy scenes removed (see text). Panel (c) shows the average TROPESS a priori CO*
*VMR for 700 to 350 hPa. Panel (d) shows the NASA Worldview SNPP/VIIRS image for 14*
*September, 2020 with clouds and smoke (true color) and fire thermal anomalies (red).*
As stated in the introduction, the TROPESS single FOV products are different from the
NUCAPS and CLIMCAPS products that combine 9 FOVs in a retrieval from a single cloud-
cleared radiance (Susskind et al., 2003). These multiple FOV products have the advantage of
increased global coverage in the presence of partially cloudy scenes but with coarser spatial
resolution. Figure 2 shows an example of CLIMCAPS (Barnet, 2019) compared to TROPESS
for SNPP/CrIS CO products (daytime only) on 13 September 2018 over the Pole Creek Fire in
Utah. For CLIMCAPS, trace gas products with less than 1 DFS report mass mixing ratio (MMR)
on a single level at the retrieval pressure with peak sensitivity, which is 500 hPa for CO. We
converted MMR to VMR for Figure 2. This is compared to the tropospheric column average
VMR from TROPESS, so the background VMR values are close, but do not represent the same
retrieved quantities. CrIS retrieval center locations are shown by the circles in Fig 2a, 2b, which
are not intended to represent the spatial extent of the observations. The CLIMCAPS retrievals
show elevated CO from the fire, but these combined FOV retrievals would give an overestimate
of the plume width and do not distinguish the larger plume from the smaller fires to the east in
Colorado.

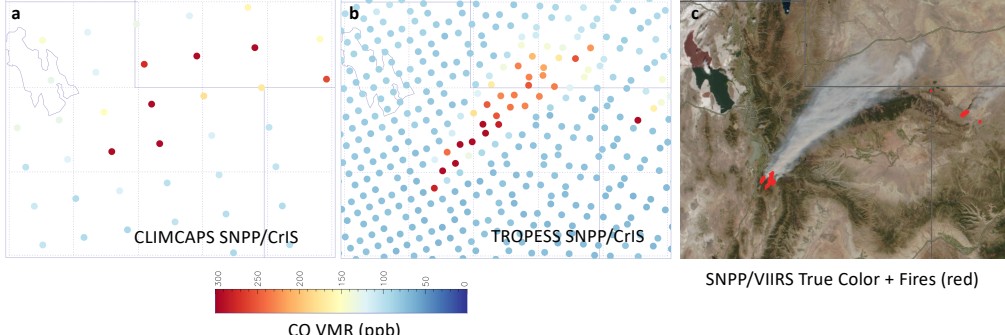

*Figure 2. Observations of the Pole Creek Fire in Utah, USA, 13 September, 2018. The Great*
*Salt Lake is in the upper left of each panel and state borders with Idaho, Wyoming and Colorado*
*are indicated by solid straight lines. Dotted lines indicate a 1° latitude by 1° longitude grid, with*
*top/left corner at 42°N, -113°E. Panel (a) shows CLIMCAPS CO at 500 hPa (MMR converted*
*to VMR). Panel (b) shows the TROPESS tropospheric CO column average VMR and panel (c)*
*shows the corresponding NASA Worldview SNPP/VIIRS image with clouds and smoke (true*
*color) and fire thermal anomalies (red).*
We note that retrievals of CO in the presence of smoke are not significantly affected by
scattering for infrared observations at wavelengths $\lambda \sim 4.6$ µm, such as in the CrIS CO band.
This is because Rayleigh scattering, which decreases by $1/\lambda^4$, is completely negligible and Mie
scattering would be significant only for particles larger than $\sim \lambda/\pi = 1.5$ µm, (e.g., Seinfeld and
Pandis, 1998), while the size distribution for biomass burning smoke particles peaks around 0.3
µm (e.g., Reid et al., 2005). For the same Pole Creek fire in Fig. 2, Juncosa Calahorrano et al.,
(2021) showed how SNPP/CrIS single pixel MUSES retrievals of acyl peroxy nitrates, also
known as PAN, along with CO, can be used to follow fire plume chemical evolution. After
subtracting background amounts, the normalized excess mixing ratios (NEMR) of PAN with
respect to CO, computed from the CrIS observations for this plume, were consistent with in situ
aircraft observations of smoke plumes from the summer 2018 WE-CAN (Western Wildfire
Experiment for Cloud Chemistry, Aerosol Absorption, and Nitrogen) campaign.
**3. Aircraft Data**



### 3.1 NOAA GML aircraft network

**3.1 NOAA GML aircraft network**
Spanning 3 decades, NOAA GML aircraft network vertical profile observations are taken on
semi-regular flights (~1/month) at fixed sites mostly in North America except for one site in
Rarotonga, Cook Islands (Sweeney et al., 2015). These flights collect air samples using an
automated flask system to obtain vertical profiles for each trace gas measured, from near the
surface to around 400 hPa, depending on aircraft limitations at each site. Flask samples are then
sent for laboratory analysis of a multitude of trace gases including CO, which was measured with
vacuum UV-fluorescence spectroscopy during the time period of this analysis. CO mixing ratios
are reported relative to the WMO X2014A scale (https://gml.noaa.gov/ccl/co_scale.html) and
have reproducibility ~1 ppb (Sweeney et al., 2015). NOAA GML aircraft profiles of CO have
been used for the long-term validation of the MOPITT CO record, with updated validation for
each new data version (Deeter et al., 2019 and references therein). For the current analysis, we
use NOAA GML aircraft network observations of CO collected during 2016 and 2017 from 7
locations (Table 1).

### 3.2 ATom aircraft campaigns

**3.2 ATom aircraft campaigns**
The Atmospheric Tomography Mission (ATom) was designed to study the most remote regions
of the Pacific and Atlantic ocean air masses in each season (Thompson et al., 2022), which also
makes the data valuable for validating satellite CO observations over a range of latitudes, with
mostly background CO concentrations, except for where transported pollution plumes were
encountered (Deeter et al., 2019; 2022; Martínez-Alonso et al., 2020; Hegarty et al., 2022). We
use CO profiles from the quantum cascade laser spectrometer (QCLS) on the ATom campaigns
1-4 (see Table 1). These NASA DC-8 flights obtained vertical profiles from 0.2 to 12 km altitude
(~290 hPa) by ascending or descending approximately every 220 km. CO was measured at 1 Hz
with QCLS reproducibility around 0.15 ppbv (McManus et al., 2010, Santoni et al., 2014). The
QCLS data were calibrated to the X2014A CO WMO scale maintained by the NOAA GML.

Table 1. Aircraft in situ validation observations used in this study.

| NOAA/GML Network flask/UV spectrometer (±1ppb CO) | | | |
|---|---|---|---|
| Code/Site name | Latitude (°N) | Longitude (°E) | Dates available |
| RTA/Raratonga | -21.25 | -159.83 | 2000-2021 |
| TGC/Offshore Corpus Christi,TX | 27.73 | - 96.86 | 2003-2021 |
| CMA/Offshore Cape May, NJ | 38.83 | - 74.32 | 2005-2022 |
| THD/Trinidad Head, CA | 41.05 | -124.15 | 2003-2022 |
| NHA/Offshore Portsmouth, NH | 42.95 | - 70.63 | 2003-2022 |
| ESP/Estevan Pt., BC | 49.38 | -128.54 | 2002-2021 |
| ACG/Alaska Coast Guard | 57.74 | -152.50 | 2009-2021 |
| | | | |
| NASA/ATom QCLS (±0.15ppb CO) | | | |
| ATom 1-4 Pacific | 75 to -65 | -150 to -70 | July 2016, Jan. 2017, Sep. 2017, April 2018 |
| ATom 1-4 Atlantic | -75 to 80 | -65 to -20 | Aug. 2016, Feb. 2017, Oct. 2017, May 2018 |

https://gml.noaa.gov/ccgg/aircraft/
https://espo.nasa.gov/atom/content/ATom





## 4. Validation Methodology

### 4.1 Data selection, coincidence criteria and vertical extension of aircraft profiles

TROPESS CrIS CO profiles are selected for comparison if they have retrieval quality of 1 and effective cloud optical depth less than 0.1 to ensure non-cloudy CrIS observations. We then find all eligible CrIS and aircraft profile pairs within 9 hours and 50 km distance. This has been a standard coincidence distance criterion for several validation studies (e.g., Deeter et al., 2019; 2022; Hegarty et al., 2022). Tang et al., (2020) found very little sensitivity in MOPITT CO validation results for 25, 50, 100 and 200 km coincidence except for the cases with a 25 km radius that resulted in an insufficient number of matches for meaningful statistics. The Tang et al. (2020) study also tested the time coincidence criterion (12, 6, 2 and 1 hour) with similar conclusions. Application of the 9 hour/ 50 km coincidence criteria yielded 2092 CrIS/aircraft profile pairs for NOAA GML flights from 2016 and 2017 and 1052 profile pairs for the ATom 1-4 campaigns. Since the aircraft profiles used for validation do not span the full vertical range of satellite retrieved profiles, we must extend these with a reasonable approximation of atmospheric CO to facilitate the comparison as described below in section 4.2. Here we use the TROPESS a priori profiles (from model climatology, described above) to extend the in situ profiles above the highest altitude sampled. The a priori profile is scaled to match the CO abundance of the aircraft measurement at the highest altitude. The choice of model and approach for extending the aircraft profiles are examined more in Tang et al., (2020) and Hegarty et al., (2022), with similar conclusions that the impacts apply mostly to bias estimates in the middle to upper troposphere. Martínez-Alonso et al., (2022) compute the uncertainty introduced by this extension explicitly using NOAA AirCore in situ balloon profiles that sample into the stratosphere (Karion et al., 2010). This uncertainty is computed for validation using aircraft profiles (with top samples around 400 hPa for NOAA/GML) by comparing MOPITT profiles to truncated and extended AirCore profiles vs. the true full AirCore profiles. The comparison error introduced by the extension was at most 3 % around 300 hPa, and much less than the standard deviation of MOPITT and full AirCore profile differences (~7-10 %) in the upper troposphere. We also note that for ATom profiles, the highest altitude samples are normally taken around 12 km (~200 hPa) and the profile extension therefore has minimal impact on tropospheric validation results.

### 4.2 Comparison of TROPESS satellite and aircraft observations

In order to account for the satellite observational and retrieval approach, including prior information, when comparing satellite retrieval products to in situ measurements of CO, we apply the instrument operator to convert the in situ profile into the values that would be retrieved for the same air mass assuming the satellite instrument and retrieval (Jones et al., 2003, Rodgers and Conner, 2003, Worden et al., 2007):

$$\hat{x}_{val} = x_a + \mathbf{A}(x_{val} - x_a) \qquad (1)$$

where $x_{val}$ is the aircraft or sonde in situ profile being used for validation (following extension, described above, and linear interpolation to the satellite vertical grid), $x_a$ is the a priori profile used in the TROPESS retrieval, $\mathbf{A}$ is the averaging kernel matrix that describes the observation and retrieval vertical sensitivity to the true state and $\hat{x}_{val}$ is the in situ validation profile transformed by the satellite instrument operator. This operation accounts for both the broad vertical resolution (or "smoothing") of remotely sensed measurements and the influence of the a



priori, which is especially important in the vertical ranges where satellite observations have low
sensitivity to CO abundance. Figure 3 shows an example of the averaging kernel **A** and a
validation comparison where Eq. 1 is applied to an ATom in situ profile.

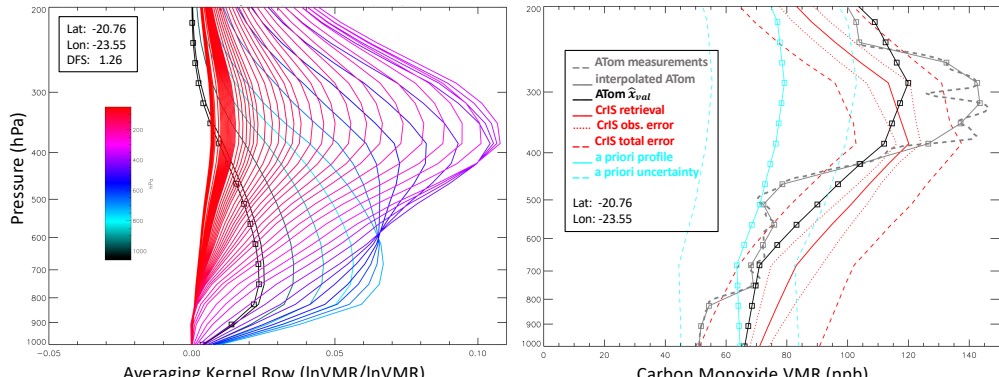


*Figure 3. Examples of TROPESS/CrIS CO averaging kernel (**A**) (left panel) and the validation
process (right panel). The colors of the averaging kernel indicate the pressure level (66 levels
from 1017.45 to 0.1 hPa) corresponding to each row, with the surface level row also indicated
by the squares. The degrees of freedom for signal (DFS), given by the sum of the diagonal (i.e.
trace) of this averaging kernel is 1.26. The right panel shows the CrIS CO profile retrieval (solid
red line) with total error (dashed red lines), observation error (dotted red lines), apriori profile
(solid cyan line with squares) and diagonal uncertainty (dashed cyan lines). The closest ATom
aircraft profile had 10.4 km; 3.5 hr coincidence. The original ATom profile (dashed grey line) is
interpolated to the CrIS vertical grid (solid grey with squares) and transformed by the
instrument operator to give ATom $\hat{x}_{val}$ (Eq. 1) (solid black line with squares).*

## 4.3 Evaluating TROPESS CO reported observational errors

Following Bowman et al., (2006, 2021), for retrieved parameter $\hat{x}$ (e.g., CO abundance) with a
priori covariance $\mathbf{S_a}$, radiance measurement covariance $\mathbf{S_e}$, Jacobian matrix $\mathbf{K} = \frac{\partial L}{\partial x}$, for radiance
$L(x)$, gain matrix $\mathbf{G} = (\mathbf{K}^T \mathbf{S_e^{-1}} \mathbf{K} + \mathbf{S_a^{-1}})^{-1} \mathbf{K}^T \mathbf{S_e^{-1}}$ and averaging kernel $\mathbf{A} = \mathbf{GK}$, the a
posteriori error covariance can be written as the sum of:

$$\mathbf{S_{\hat{x}}} = \mathbf{S_{smoothing}} + \mathbf{S_{observational}} \qquad (2)$$

with $\mathbf{S_{smoothing}} = (\mathbf{I} - \mathbf{A_{xx}})\mathbf{S_a}(\mathbf{I} - \mathbf{A_{xx}})^{\mathbf{T}}$ and

$$\mathbf{S_{observational}} = \mathbf{S_{noise}} + \mathbf{S_{cross-state}} + \mathbf{S_{systematic}} \qquad (3)$$

where $\mathbf{S_{noise}} = \mathbf{GS_e G^T}$ , $\mathbf{S_{cross-state}} = \sum_{b\_ret} \mathbf{A_{xs}} \mathbf{S_a^{b\_ret}} \mathbf{A_{xs}}^{\mathbf{T}}$ and

$$\mathbf{S_{systematic}} = \sum_{b} \mathbf{GK_b S_b (GK_b)^{T}} \qquad (4)$$

In this notation, *b* variables are parameters that are held constant in the retrieval but affect the
radiance observation used for the CO retrieval through Jacobian $\mathbf{K_b}$ while *b_ret* variables are





314 retrieved along with CO and have corresponding off-diagonal terms in the full retrieval
315 averaging kernel matrix. When we apply the satellite instrument operator in Eq. 1 to the in situ
316 aircraft profile, we are accounting for the smoothing error term. Thus, we expect differences
317 between $\hat{x}_{val}$ and our retrieved $\hat{x}$ to be due to observational error terms (Eq. 3) and to
318 geophysical differences from the sampling of different airmasses and surface locations because
319 of imperfect coincidence.
320

321 **5. Validation Results**
322

323 **5.1 TROPESS CrIS CO comparisons with NOAA GML**
324 After extending the in situ profiles vertically (described in Sec. 4.1) and applying Eq. 1, we
325 compute the differences between satellite retrievals and transformed aircraft profiles. Figure 4
326 shows the bias (% relative difference) of the CrIS CO retrieved profiles with respect to NOAA
327 GML aircraft profiles ($\hat{x}_{val}$). A similar pattern of positive bias in the lower to mid troposphere
328 and negative bias in the upper troposphere is observed for MUSES-AIRS profiles compared to
329 NOAA GML flights (Hegarty et al., 2022). However, MOPITT (version 9, TIR-only data)
330 comparisons to NOAA GML (Deeter et al., 2022) have almost the opposite vertical bias pattern
331 with a negative bias ( -1.6 %) in the lower to middle troposphere and a positive bias (0.6 %) in
332 the upper troposphere. Table 2 gives the mean bias and standard deviations for selected pressures
333 and partial column average VMR over different observing conditions (land, ocean, day and
334 night). The partial column refers to the CO column between the minimum and maximum flight
335 altitudes of each aircraft profile. The average VMR over this range is computed by interpolating
336 both the CrIS retrieval and the aircraft $\hat{x}_{val}$ profile to these endpoints. Since aircraft flights
337 normally occur during daytime, there are fewer coincident pairs for CrIS night retrievals. Tang et
338 al. (2020) find larger bias and variance for nighttime MOPITT data in comparisons with in situ
339 aircraft data, especially for flights over urban regions, suggesting more night validation flights
340 are needed to properly evaluate night satellite retrievals.





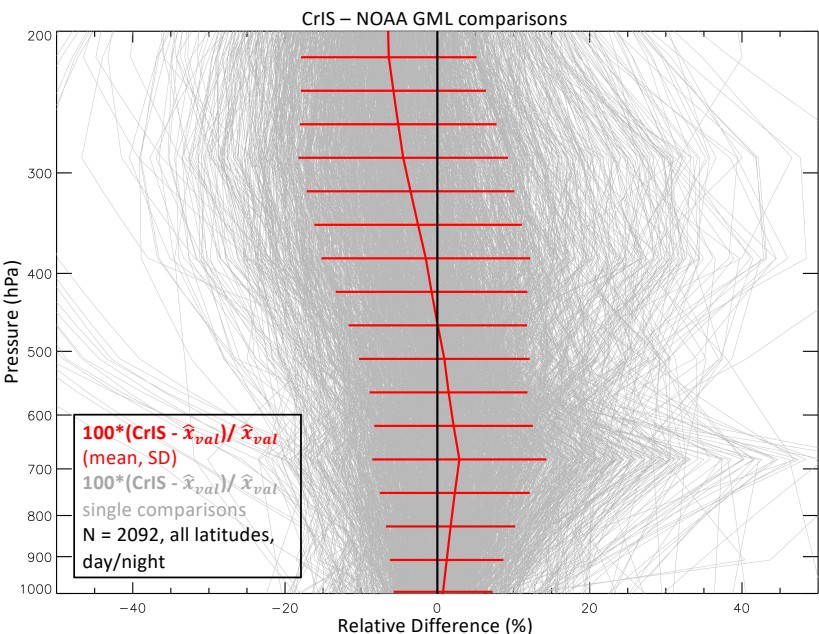


**Figure 4.** *Relative differences ( %) in single CrIS retrievals with coincident NOAA GML $\hat{x}_{val}$*
*profiles (grey) and the average % difference with $1\sigma$ horizontal bars (red). Both day and night*
*CrIS observations are included for coincidence search with 1866 day and 266 night comparison*
*pairs found.*
Table 2. Bias and standard deviation (SD) for comparisons of TROPESS SNPP/CrIS CO
retrievals and in situ CO profiles from NOAA GML fights.

| Obs. type | % bias 750 hPa | % SD 750 hPa | % bias 511 hPa | % SD 511 hPa | % bias 287 hPa | % SD 287 hPa | % bias Column | % SD Column | # pairs |
|---|---|---|---|---|---|---|---|---|---|
| All | 2.29 | 9.84 | 0.92 | 11.20 | -4.48 | 13.76 | 0.57 | 8.56 | 2092 |
| Land | 3.04 | 10.85 | -0.044 | 11.95 | -6.15 | 13.97 | 1.24 | 9.46 | 853 |
| Ocn | 1.78 | 9.04 | 1.58 | 10.59 | -3.33 | 13.49 | 0.11 | 7.84 | 1239 |
| Day | 1.97 | 9.79 | 0.13 | 10.93 | -5.37 | 13.32 | 0.23 | 8.77 | 1866 |
| Ngt | 4.94 | 9.86 | 7.36 | 11.27 | 2.81 | 15.05 | 3.41 | 5.82 | 266 |

Figure 5 shows how the observed partial column average VMR and CrIS retrieval bias with
respect to NOAA GML $\hat{x}_{val}$ profiles vary with latitude and Figure 6 shows how these vary with
time. No significant bias dependence on latitude is observed for the NOAA GML flight sites.
Although a bias drift of $-0.007 \pm 0.001$ %/day is detected, we recognize that our comparison
time range is not sufficient for a reliable estimate of bias drift, and more years of comparisons
would be required.



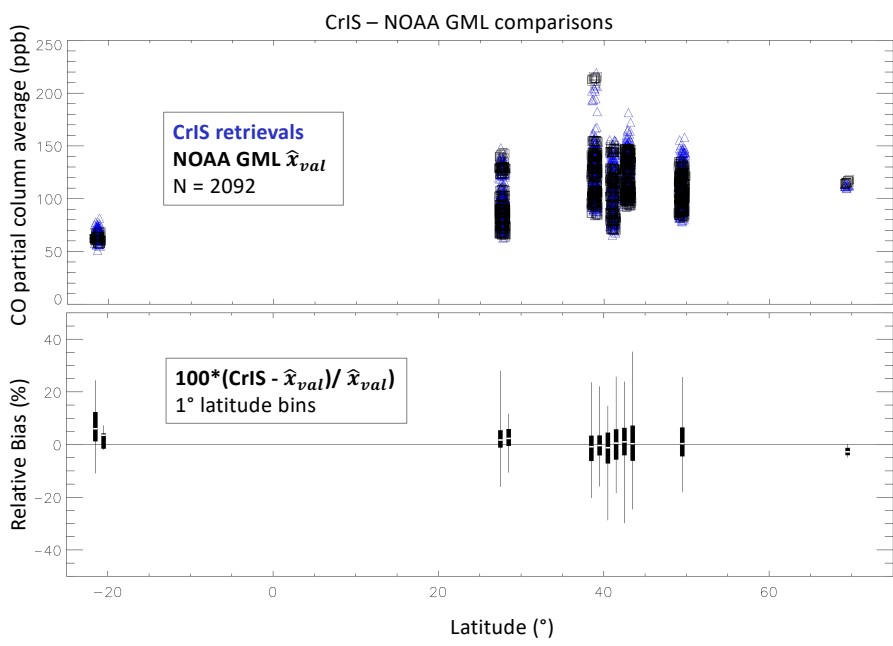

**Figure 5.** *Latitude dependence of CO partial column average VMR (ppb) for TROPESS CrIS retrievals and NOAA GML $\hat{x}_{val}$ (upper panel) and bias difference statistics (lower panel) shown by box/whisker symbols representing minimum and maximum values (whisker), lower quartile (box bottom), median (white stripe), and upper quartile (box top). A minimum of 5 comparisons per bin was required.*

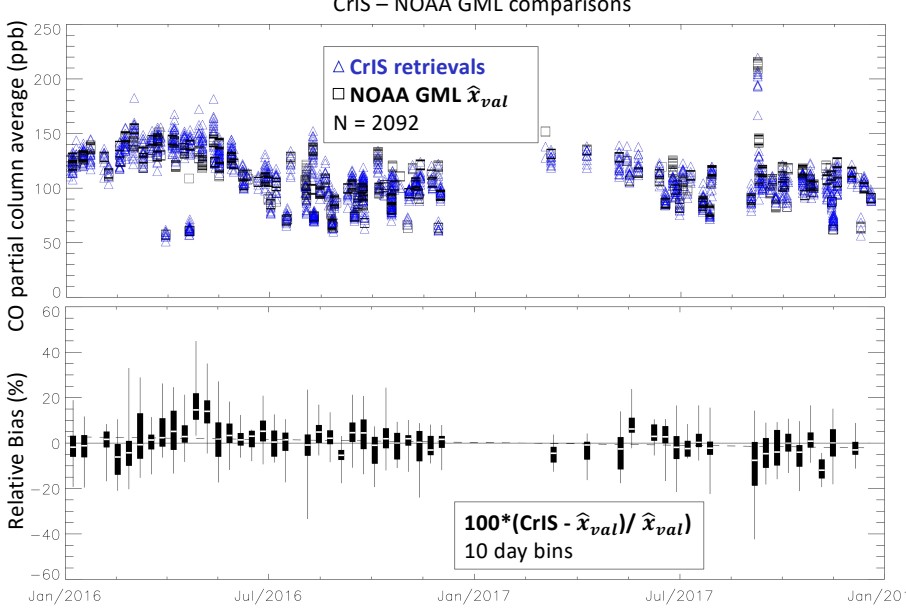




*Figure 6. Time dependence of CO partial column average VMR (ppb) for TROPESS CrIS*
*retrievals and NOAA GML $\hat{x}_{val}$ (upper panel) and bias difference statistics (lower panel) shown*
*by box/whisker symbols representing minimum and maximum values (whisker), lower quartile*
*(box bottom), median (white stripe), and upper quartile (box top). A minimum of 5 comparisons*
*per bin was required. The dashed line indicates a fit for bias drift (see text).*
**5.2 TROPESS CrIS CO validation with ATom**
Figure 7 shows the bias ( % relative difference) of the CrIS CO retrieved profiles with respect to
ATom $\hat{x}_{val}$ in situ profiles for all latitudes and 3 latitude ranges: 30°S to 30°N, 90°S to 30°S, and
30°N to 90°N. The vertical behavior of the bias is similar to the above CrIS comparisons with
NOAA GML flights, with positive bias in the lower troposphere and negative bias in the upper
troposphere and is also similar to the MUSES-AIRS CO profiles compared to ATom flights
(Hegarty et al., 2022). However, for MOPITT V9T comparisons to ATom flights (Deeter et al.,
2022), the vertical bias pattern is again mostly opposite, with a negative bias (~4 %) in the lower
to mid troposphere and a positive bias (~2 %) in the upper troposphere. CrIS CO comparisons
with ATom have less variance than comparisons with NOAA GML, especially for 90°S to 30°S.
Table 3 gives the mean bias and standard deviations for selected pressures and partial column
average VMR over different observing conditions (land, ocean, day and night) and latitude
ranges. As described above, the partial column average VMR is computed over the altitude
ranges of each aircraft profile. Due to the nature of the ATom campaign, there are fewer
observations over land.

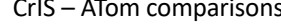

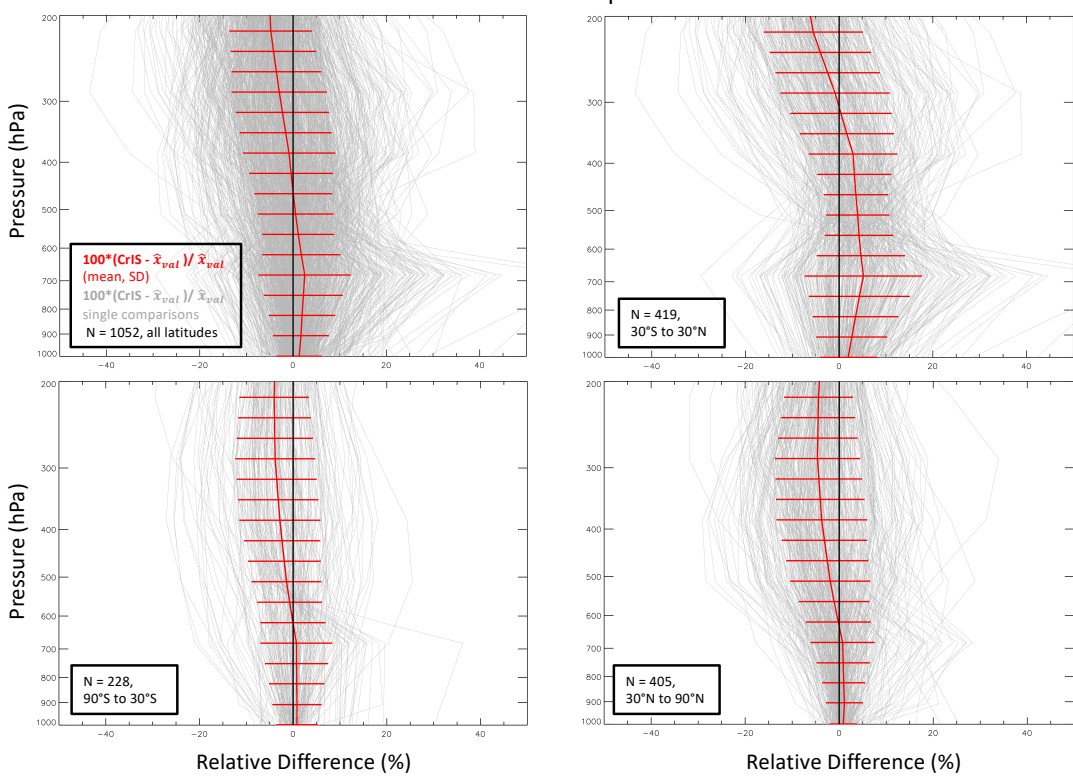



**Figure 7**. *Relative differences (%) in single CrIS retrievals with coincident ATom $\hat{x}_{val}$ profiles*
*(grey) and the average % difference with $1\sigma$ horizontal bars (red). Latitude ranges are*
*indicated in each panel along with the number of comparison pairs. Both day and night CrIS*
*observations are included.*
Figure 8 shows how the observed partial column average VMRs and CrIS retrieval bias with
respect to ATom $\hat{x}_{val}$ profiles vary with latitude. It appears that tropical and northern hemisphere
sub-tropical latitude ranges have a slightly higher positive bias than what is observed for higher
latitudes, potentially indicating a TROPESS CrIS retrieval issue with water vapor or some other
interferent that is not fully characterized and requires further investigation.
Table 3. Bias and standard deviation (SD) for comparisons of TROPESS SNPP/CrIS CO
retrievals and in situ CO profiles from ATom flight campaigns 1-4.

| Obs. type | Latitude Range (°) | % bias 750 hPa | % SD 750 hPa | % bias 511 hPa | % SD 511 hPa | % bias 287 hPa | % SD 287 hPa | % bias Col. | % SD Col. | # pairs |
|---|---|---|---|---|---|---|---|---|---|---|
| All | all | 2.21 | 8.46 | 0.54 | 8.12 | -2.95 | 10.24 | -0.035 | 5.91 | 1052 |
| Land | all | 1.20 | 4.15 | -0.49 | 7.59 | -2.95 | 10.46 | -0.79 | 7.09 | 102 |
| Land | 30S-30N | - | - | - | - | - | - | - | - | 1 |
| Land | 30N-90N | 1.22 | 4.27 | -0.69 | 7.76 | -3.25 | 10.70 | -0.91 | 7.32 | 95 |
| Land | 90S-30S | 0.12 | 0.29 | 0.89 | 2.35 | 1.84 | 4.65 | 0.67 | 1.86 | 6 |
| Ocn | all | 2.32 | 8.79 | 0.65 | 8.17 | -2.95 | 10.21 | 0.046 | 5.76 | 950 |
| Ocn | 30S-30N | 4.32 | 10.80 | 3.96 | 6.75 | -0.86 | 11.67 | 2.33 | 5.44 | 418 |
| Ocn | 30N-90N | 0.75 | 6.01 | -2.28 | 8.70 | -5.03 | 8.51 | -2.22 | 6.34 | 310 |
| Ocn | 90S-30S | 0.74 | 6.85 | -1.46 | 7.5 | -3.98 | 8.57 | -1.09 | 3.49 | 222 |
| Day | all | 2.62 | 8.76 | 0.53 | 7.91 | -3.21 | 9.81 | 0.010 | 5.85 | 782 |
| Day | 30S-30N | 4.94 | 11.42 | 3.55 | 6.57 | -2.01 | 10.99 | 2.23 | 5.16 | 300 |
| Day | 30N-90N | 0.91 | 5.76 | -1.63 | 8.62 | -4.33 | 9.22 | -1.68 | 6.74 | 331 |
| Day | 90S-30S | 1.79 | 6.90 | -0.72 | 6.71 | -3.11 | 8.12 | -0.70 | 2.91 | 151 |
| Ngt | all | 1.03 | 7.39 | 0.57 | 8.71 | -2.21 | 11.36 | -0.17 | 6.08 | 270 |
| Ngt | 30S-30N | 2.79 | 8.82 | 5.02 | 7.07 | 2.03 | 12.73 | 2.59 | 6.09 | 119 |
| Ngt | 30N-90N | 0.68 | 5.15 | -3.16 | 7.93 | -5.88 | 8.45 | -2.98 | 5.84 | 74 |
| Ngt | 90S-30S | -1.35 | 5.94 | -2.73 | 8.58 | -5.25 | 9.15 | -1.73 | 4.30 | 77 |




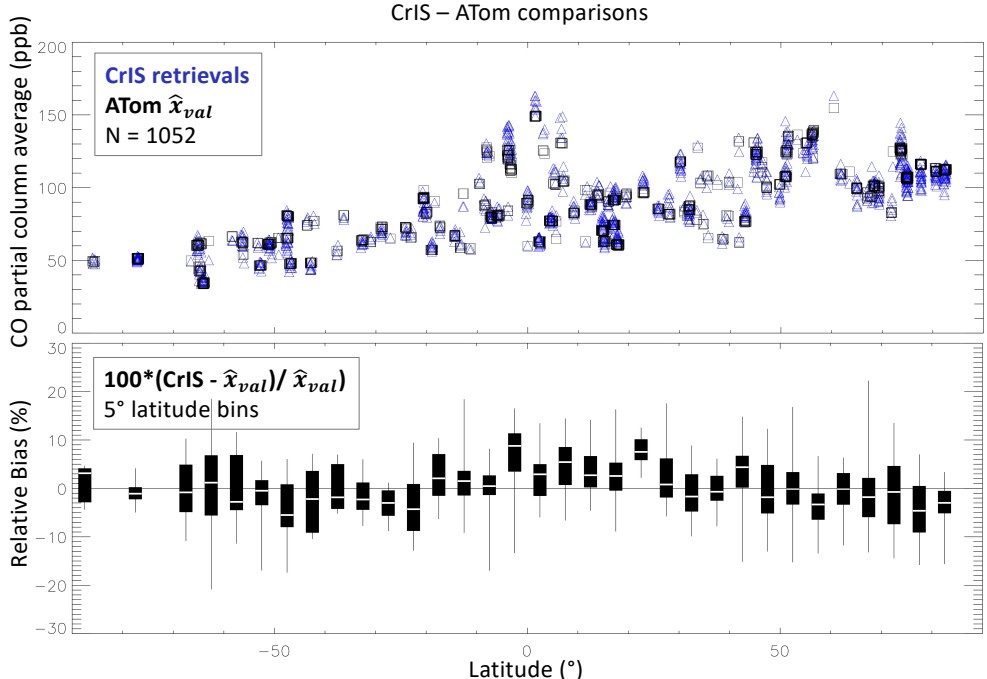

***Figure 8**. Latitude dependence of CO partial column average VMR (ppb) for TROPESS CrIS*
*retrievals and ATom $\hat{x}_{val}$ (upper panel) and bias difference statistics (lower panel) shown by*
*box/whisker symbols representing minimum and maximum values (whisker), lower quartile (box*
*bottom), median (white stripe), and upper quartile (box top). A minimum of 5 comparisons per*
*bin was required.*
In Figure 9, we examine the seasonal behavior of CO sampled by ATom and CrIS in mostly
remote ocean regions. In the high latitude southern hemisphere (SH), we see the lowest values in
summer and fall (Jan/Feb and Apr/May) as expected due to the chemical destruction of CO in a
region with few local combustion sources. In the tropics, we find high values corresponding to
African and South American biomass burning plumes over the Atlantic in all seasons except
Northern Hemisphere (NH) spring. Lower values of CO in the tropics for NH summer and winter
correspond to profiles over the Pacific ocean (e.g., Strode et al., 2018, Bourgeois et al., 2020).
The close alignment of the CrIS and ATom $\hat{x}_{val}$ partial column average values in Fig. 9 indicates
that CrIS is able to capture the seasonal, latitudinal and hemispherical variations observed by
ATom.



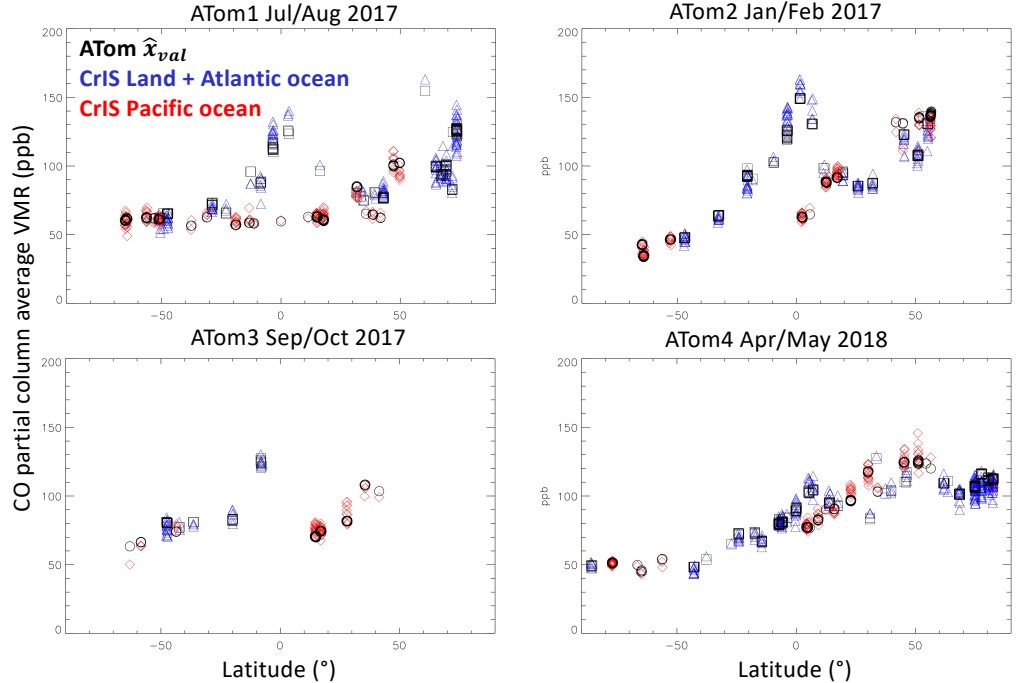

**Figure 9**. *Latitude dependence of partial column average CO for each ATom campaign. Black squares ATom $\hat{x}_{val}$ partial column average values over Atlantic Ocean scenes; black circles indicate ATom values over Pacific Ocean scenes. Blue triangles indicate CrIS CO partial column average values over land and Atlantic Ocean scenes; red diamonds indicate CrIS values over Pacific Ocean scenes.*

**5.3 Dependence on CO amount**

For both the NOAA GML and ATom flights we find a small negative dependence of TROPESS CrIS retrieval bias with respect to CO amount, with magnitude less than 0.1 %/ppb. Figure 10 shows how the partial column average VMR bias varies with CO VMR for the two validation data sources and we can also see how ATom flights sampled air with lower CO concentrations. Fig. 10 indicates that TROPESS CrIS CO average column VMRs have very little dependence on CO amount and we find similar results for CrIS retrieved CO at vertical levels 511 hPa and 750 hPa (shown in the supplementary material).



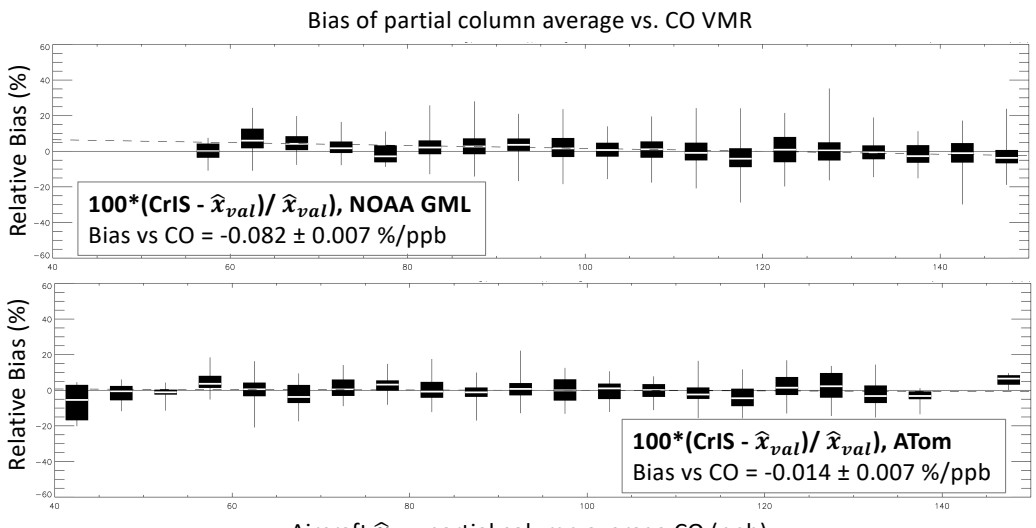

**Figure 10.** *Bias of CrIS partial column average CO vs CO amount for NOAA GML flights in top*
*panel and ATom flights in bottom panel with box/whisker symbols in 5 ppb bins. Linear*
*regression results are shown in the legend boxes.*
**5.4 Evaluation of TROPESS CrIS CO retrieval observational errors**
Here we compare the observed variance of differences between retrieved CrIS CO profiles and in
situ aircraft profiles, after applying Eq. 1, with the TROPESS reported observational errors
defined in Eqs. 3 and 4. As described in section 4.3, we expect the differences between retrieved
CrIS and aircraft CO profiles ($\hat{x}_{val}$) to have a variance due to the combination of observational
errors and geophysical variation from imperfect coincidence. Figure 11 shows comparisons of
individual and average computed observational fractional errors to the standard deviation (SD) of
CrIS - $\hat{x}_{val}$ profile differences as well as the diagonal for the a priori covariance and the SD of
prior - $\hat{x}_{val}$ profile differences As expected, the average observational errors are less than
SD(CrIS - $\hat{x}_{val}$), but in some vertical ranges, they are much less and could be underestimated via
instrument and systematic error assumptions in the TROPESS retrieval as Hegarty et al., (2022)
suggest. Additional studies to test the sensitivity of the comparison variance to a range of
coincidence criteria are needed to confirm a retrieval underestimate, but these would require
several repeated validation measurements for the same observing conditions.
Despite the potential for underestimated observational errors, the general behavior of the error
comparison is what we expect for an optimal estimation retrieval and we can see the retrieval
influence on the shape of SD(CrIS - $\hat{x}_{val}$). Near the surface, where there is less retrieval
sensitivity as indicated by the averaging kernel, we see that SD(prior - $\hat{x}_{val}$) becomes smaller
than SD(CrIS - $\hat{x}_{val}$). This is expected from Eq. 1 since the priori contribution becomes more
dominant in $\hat{x}_{val}$ for vertical ranges with less retrieval sensitivity. In contrast, for the middle
troposphere where we have the most sensitivity for TIR remote sensing, it is clear that SD(CrIS -
$\hat{x}_{val}$) represents an improvement over SD(prior - $\hat{x}_{val}$). In Figure 12, the error comparison is
shown separately for 3 ATom latitude ranges and we can see that the agreement between





observational errors and SD(CrIS - $\hat{x}_{val}$) is closest for ATom flights in the mostly clean middle
to high latitude southern hemisphere, where it is most likely that the aircraft and satellite are
observing similar airmasses with background CO concentrations. These results give confidence
that TROPESS single retrieval error characterization can be used to weight data for averaging
and inverse analysis applications.

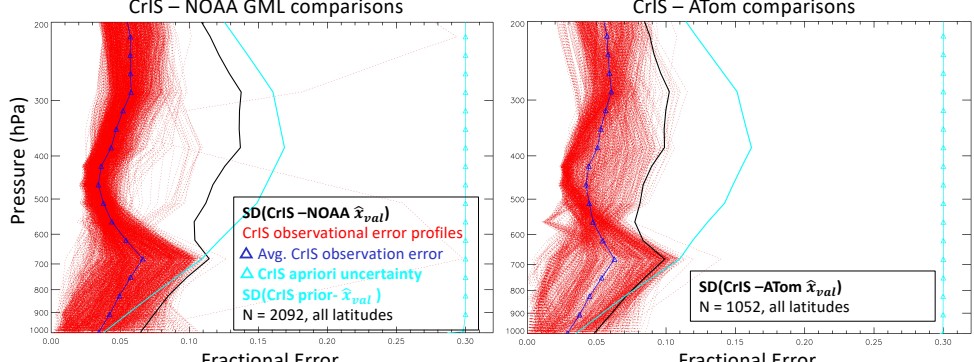

***Figure 11***. *Error comparison of CrIS observational error estimates and the standard deviation*
*(SD) of CrIS-* $\hat{x}_{val}$ *(in black) for NOAA GML flights in the left panel and ATom flights in the right panel.*
*Single profile CrIS observational error estimates are plotted in red, with average in dark blue with*
*triangles. For reference, and the standard deviation of CrIS prior with aircraft* $\hat{x}_{val}$ *is in cyan and the*
*apriori fractional uncertainty (0.3) is shown in cyan with triangles.*

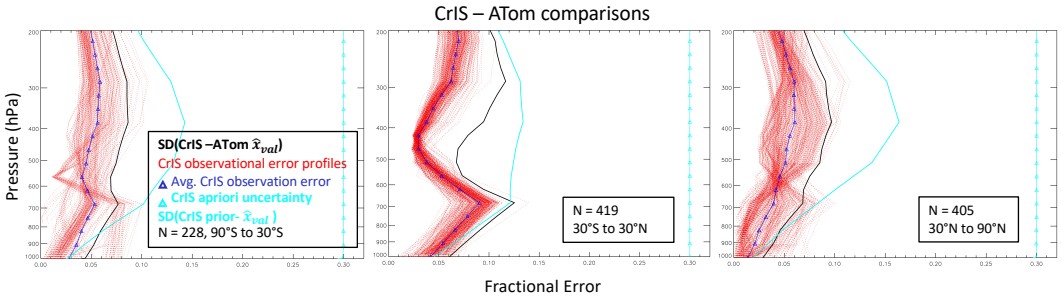

***Figure 12***. *Same as Fig. 11 but for 3 ATom latitude ranges.*
**6. Summary and Conclusions**
This study used in situ observations from routine NOAA GML flights and the four ATom
campaigns to evaluate TROPESS single pixel CO retrievals from the SNPP/CrIS FTS
instrument. We find that:

1) The single FOV CrIS product provides improved representation of CO in smoke plumes compared to retrievals that combine multiple FOVs.
2) Comparisons with aircraft in situ profiles (after extension, interpolation and application of Eq. 1) show that biases have a vertical dependence in the troposphere that is consistent for both sets of in situ data with average biases that are positive (~ 2.3 %) in the lower troposphere and negative (~ -4.5 %) in the upper troposphere.
3) Small biases (0.6 % and -0.04 % for NOAA GML and ATom, respectively) are observed for the CrIS CO partial column average VMR corresponding to the aircraft profile vertical ranges.
4) No significant latitude dependence of CrIS CO column bias is found for the NOAA GML comparisons, but comparisons with ATom, which better covered a range of latitudes, have a slightly more positive bias for tropical scenes that could indicate a small, uncharacterized retrieval dependence on water vapor or another interferent species.
5) CrIS CO retrievals capture the seasonal and spatial variations observed by ATom.
6) There is a small negative dependence (magnitude < 0.1 %/ppb) of CrIS bias on CO amount.
7) Comparisons of computed observational errors and standard deviations of retrieval-aircraft comparison differences show the expected behavior for optimal estimation retrievals and demonstrate improvement over the standard deviation of prior-aircraft differences.

TROPESS CrIS CO biases detected in this study are in general much smaller than comparison standard deviations. We therefore make no recommendations for automated bias corrections in data processing, similar to other validation studies for satellite CO retrievals (e.g., Deeter et al, 2019; 2022). This is unlike other TROPESS products such as $CH_4$ (Kulawik et al., 2021) where a bias correction is more appropriate given the size of bias detected as well as the atmospheric lifetime (~10 years for methane) and reduced atmospheric variability compared to CO. Each analysis using TROPESS CrIS CO data must consider the variability of CO over the domain of interest and ascertain whether the biases observed here could affect numerical conclusions. The biases reported from this study will need to be included when long term records of satellite CO observations are harmonized and used together for computing trends, data assimilation or other analyses. For example, with the 22-year record of MOPITT CO profiles, this is especially important when combining datasets since the vertical bias pattern for MOPITT data with respect to in situ observations has a positive bias in the upper troposphere and negative bias in the lower to middle troposphere with the opposite behavior compared to the TROPESS/CrIS vertical bias pattern.

Future validation of the TROPESS CrIS CO products will include a longer time record of comparisons and quantification of bias drift, for CrIS on SNPP and on the JPSS satellite series. The validation results presented here demonstrate that these products are suitable for tropospheric CO data analyses. The bias at all vertical levels is <10 % and error characterization for single retrievals can be used to weight data for averaging and applications such as data assimilation and inverse modelling.



*Data availability.* The NOAA GML data were obtained from https://doi.org/10.7289/V5N58JMF (Sweeney et al. 2021). The ATom aircraft data were obtained from https://doi.org/10.3334/ORNLDAAC/1581 (Wofsy et al., 2018). CrIS MUSES CO products are available via the GES DISC from the NASA TRopospheric Ozone and its Precursors from Earth System Sounding (TROPESS) project at https://doi.org/10.5067/I1NONOEPXLHS (Bowman, 2021). The CrIS–aircraft matched data set used here for validation is available from the authors on request.

*Author contributions.* HMW, GLF, SSK, JDH, KCP, ML and VHP designed the study and HMW prepared the manuscript. GLF analyzed the satellite/aircraft comparisons and prepared the figures, SSK, KB, DF, VK, ML, KCP, VHP, JRW developed the MUSES algorithm and provided the CrIS CO retrievals. RC and KM participated in the ATom campaign and provided guidance in the use of the measurements. KM provided the NOAA GML aircraft data. All authors reviewed and edited the manuscript.

*Competing Interests.* Some authors are members of the editorial board of AMT. The peer-review process was guided by an independent editor, and the authors have no other competing interests to declare.

*Acknowledgements.* This research was conducted at the National Center for Atmospheric Research (NCAR), which is sponsored by the National Science Foundation. Part of this research was carried out at the Jet Propulsion Laboratory (JPL), California Institute of Technology, under a contract with the National Aeronautics and Space Administration. The NOAA GML aircraft observations are supported by NOAA and CIRES. The ATom aircraft data were supported by the NASA Airborne Science Program and Earth Science Project Office. We acknowledge the use of imagery from the NASA Worldview application (*https://worldview.earthdata.nasa.gov/*), part of the NASA Earth Observing System Data and Information System (EOSDIS). We thank Dr. Benjamin Gaubert for his NCAR internal review of the manuscript.

*Financial support.* The Jet Propulsion Laboratory (JPL), California Institute of Technology, is under a contract with the National Aeronautics and Space Administration (80NM0018D0004). This research has also been supported by NASA via the TRopospheric Ozone and its Precursors from Earth System Sounding (TROPESS) project at JPL and a NASA ROSES award: 80NSSC18K0687. The NOAA Cooperative Agreement with CIRES is NA17OAR4320101. The NCAR facility is sponsored by the National Science Foundation (grant no. 1852977).

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
