# Peer review of "TROPESS/CrIS carbon monoxide profile validation with NOAA GML and ATom in situ aircraft observations"

_Atmospheric Measurement Techniques, 2022_

## Referee Comment (RC1)

General comments:

The authors present a study on the validation of TROPESS/CrIS carbon monoxide profiles. These TROPESS CrIS data retrieved using the MUSES algorithm with single field of view (FOV) radiances provide a better spatial resolution and allows to study plumes in more detail. Therefore, these CO profiles are very valuable when properly validated.

In this paper, this data set is validated against in-situ data from aircraft observations. Averaging kernels are applied to take into account different vertical resolutions. The retrieved CO profiles agree well with the in-situ profiles.

Therefore, I would recommend publishing this paper after minor revisions. The paper is well-written and fits well to the scope of AMT. Please also see specific comments below.

Specific comments:

- p. 7, line 238: Please provide a definition of 'retrieval quality of 1'.

- p. 10: Is there a reason for limiting the study to 2 years of data?

- p. 13&14: line 396 indicates a potential issue with water vapor: 'potentially indicating a TROPESS CrIS retrieval issue with water vapour or some other interferent'. On the other hand, Fig. 9 and lines 417 to 420 states the seasonal variations are well captured. In case of an $H_2O$ retrieval issue a seasonal variation of the difference between remote sensing and in-situ product is expected, at least outside the tropics. Can you elaborate a bit more on this and the seasonal dependence of the difference between TROPESS CrIS and in-situ data?

- p. 18: I missed a comparison with validation results using different retrieval approaches, for example with the multiple FOVs retrieval.

Technical corrections:

- p. 1: TROPESS/CrIS in the title, TROPESS CrIS later in the text
- p. 3, line 101: TROPOESS => TROPESS
- p. 5, Fig. 2: Some lines are hard to see
- p. 8, Fig. 3: Axis scale is hard to read
- p. 9, line 323: I would suggest to add 'aircraft data' or similar:
'TROPESS CrIS CO comparisons with NOAA GML' =>
'TROPESS CrIS CO comparisons with NOAA GML aircraft data'
- l. 197: Calahorranol et al: 2018 => 2021
- l. 767 McMillam => McMillan
- l. 876: a blank line is missing

---

## Referee Comment (RC2)

**General comments**

This study presents the results of validating TROPESS/CrIS carbon monoxide (CO) profiles against in-situ aircraft observations. The authors demonstrate the significance of high-resolution spaceborne CO observations and present a data quality assessment. In particular, Section 6 assists readers in how interpreting the TROPESS/CrIS data in detail. This information is essential for users applying this data to their studies.

The manuscript fits the scope of the journal and is well written. I believe it can be more readable with a bit of clarification and elaboration. Therefore, I would recommend publishing this manuscript after minor revisions. Please see specific comments below.

**Specific comments**

- Section 1 (Introduction):
  In general, more clarifications would be appreciated on the definitions of MUSES and TROPESS. Although detailed information is given in Section 2, please consider elaborating briefly on the description of the two terms (i.e., MUSES and TROPICS) in Section 1. My main questions are: Is it correct that MUSES is the algorithm name and TROPESS is the product name?
    - More specifically, on Lines 84–87:
      (1) Please provide full names for the abbreviations MUSES and TROPESS here. MUSES' full name is provided in Line 91, after its first appearance in Line 84. TROPESS' full name is only given in the abstract or after the conclusion section. (2) Please consider moving a part (or the entirety) of the descriptions of MUSES (Lines 99–106) and TROPESS (Lines 94–98) above Line 84.

- Line 238: Please provide a definition of the quality flag used here.

- Lines 329–332 and 378–380: The fact that MOPITT shows different patterns is mentioned twice in this manuscript without describing the causes. Could you provide possible reasons (e.g., differences in instruments or algorithms)?

- Lines 458 and 507: The phrases "what we expect for an optimal estimation" (Line 458) and "the expected behavior for optimal estimation retrievals" (Line 507) sound vague. Please elaborate on these sentences.

**Technical corrections**

- Line 78: CO2 → $CO_2$

- Lines 85, 197, and 452: Please remove a comma before a parenthesis when presenting a reference (e.g., Hegarty et al., (2022) → Hegarty et al. (2022)).

- Line 147: x → ×

- Line 188 and Table 1: How about replacing °E with °W since all longitude values are negative?

- Line 433: Fig. 10 → Figure 10

- Line 450: A period missing (differences As expected → differences. As expected)

---

## Author Comment (AC1)

Please find our responses embedded below in *blue italics.*

Best regards,
Helen Worden & co-authors

Reviews of amt-2022-128
"TROPESS/CrIS carbon monoxide profile validation with NOAA GML and ATom in situ aircraft observations"

Anonymous Referee #1

General comments: The authors present a study on the validation of TROPESS/CrIS carbon monoxide profiles. These TROPESS CrIS data retrieved using the MUSES algorithm with single field of view (FOV) radiances provide a better spatial resolution and allows to study plumes in more detail. Therefore, these CO profiles are very valuable when properly validated. In this paper, this data set is validated against in-situ data from aircraft observations. Averaging kernels are applied to take into account different vertical resolutions. The retrieved CO profiles agree well with the in-situ profiles. Therefore, I would recommend publishing this paper after minor revisions. The paper is well written and fits well to the scope of AMT. Please also see specific comments below.

*Response:*
*We thank the referee for their time and effort to review the paper and valuable comments that have helped to improve the manuscript.*

Specific comments:
- p. 7, line 238: Please provide a definition of 'retrieval quality of 1'.

*Response:*
*We have added the following paragraph to Sec. 2.1 (TROPESS retrieval approach):*
*The TROPESS CO products have quality flags for screening cases that did not converge or that have unphysical results. This screening checks the magnitude and spectral structure of radiance residuals, cloud retrieval characteristics, and deviation of surface emissivity from a priori values. Specifically, retrievals with good data quality of 1 have:  radiance residual standard deviation less than 12 times the radiance error, an absolute value of the radiance residual mean less than 0.7 times the radiance error, KdotDL (the normalized dot product of the Jacobians and the radiance residual) less than 0.8, LdotDL (the normalized dot product of the radiance and the residual) less than 0.6, cloud top pressures below 90 hPa, mean cloud optical depths less than 50, cloud variability (variation with respect to wavenumber) less than 3, and mean surface emissivity that did not change by more than 0.06. These threshold values are based on comparisons with in situ data and other satellite data to determine when retrievals are valid.*

- p. 10: Is there a reason for limiting the study to 2 years of data?
*Response:*
*This was due to the logistics and priorities of data processing. We decided to proceed with the study using only the 2 years that span both ATom and NOAA flights in order to make validation results available for this unique data set. Further validation over a longer time range and*

*extending the analysis to NOAA-20/CrIS will be the topics of future studies, as stated in the conclusions.*

- p. 13&14: line 396 indicates a potential issue with water vapor: 'potentially indicating a TROPESS CrIS retrieval issue with water vapour or some other interferent'. On the other hand, Fig. 9 and lines 417 to 420 states the seasonal variations are well captured. In case of an H2O retrieval issue a seasonal variation of the difference between remote sensing and in-situ product is expected, at least outside the tropics. Can you elaborate a bit more on this and the seasonal dependence of the difference between TROPESS CrIS and in-situ data?

*Response:*
*Since this bias latitude dependence is barely detectable, it is not likely that we have enough ATom coincidences by season to see the same effect that we see in the tropics for all data, so seasonal water vapor dependencies outside of the tropics will need to be studied more with the NOAA GML observations and more years of CrIS retrievals. The bias in the tropics could be similar to the water vapor dependence found for MOPITT (Deeter et al., 2019), but we will also need to consider the possible interference of $N_2O$ (Gonzalez et al., 2021) when investigating this slightly higher bias. We have added more detail to this paragraph on the possible interferents that could contribute to a bias:*

For example, Deeter et al. (2018) found that an empirical correction to MOPITT radiances resulting from a linear dependence on water vapor removed most of the latitude dependent bias in MOPITT CO profiles. Another gas interferent in the TIR CO band is $N_2O$ and we will also need to consider the latitude dependent $N_2O$ anomalies observed by ATom (Gonzalez et al., 2021) when assessing the contributions to this latitude dependence in TROPESS/CrIS CO bias.

*Adding the reference:*
Gonzalez, Y., Commane, R., Manninen, E., Daube, B. C., Schiferl, L. D., McManus, J. B., McKain, K., Hintsa, E. J., Elkins, J. W., Montzka, S. A., Sweeney, C., Moore, F., Jimenez, J. L., Campuzano Jost, P., Ryerson, T. B., Bourgeois, I., Peischl, J., Thompson, C. R., Ray, E., Wennberg, P. O., Crounse, J., Kim, M., Allen, H. M., Newman, P. A., Stephens, B. B., Apel, E. C., Hornbrook, R. S., Nault, B. A., Morgan, E., and Wofsy, S. C.: Impact of stratospheric air and surface emissions on tropospheric nitrous oxide during ATom, Atmos. Chem. Phys., 21, 11113–11132, https://doi.org/10.5194/acp-21-11113-2021, 2021.

- p. 18: I missed a comparison with validation results using different retrieval approaches, for example with the multiple FOVs retrieval.

*Response:*
*Thank you for pointing out this oversight. We found a reference for NUCAPS/CrIS CO profile validation by Nalli et al. (2020). We thought the most appropriate place for a comparison with the reference for multiple FOV retrieval validation was in section 5.2 since Nalli et al. (2020) describe comparisons of the NUCAPS CO profiles with ATom in situ data. We now include the following text:*

This TROPESS/CrIS CO bias also differs from Nalli et al. (2020) who examined the bias of NUCAPS profiles (including CO) with respect to ATom in situ profiles. That study, using the multiple FOV NUCAPS retrievals, found a small positive bias (~2%) for SNPP/CrIS CO with respect to ATom CO at all tropospheric vertical levels after applying their averaging kernels.

*Adding the reference:*
Nalli, N.R.; Tan, C.; Warner, J.; Divakarla, M.; Gambacorta, A.; Wilson, M.; Zhu, T.; Wang, T.; Wei, Z.; Pryor, K.; Kalluri, S.; Zhou, L.; Sweeney, C.; Baier, B.C.; McKain, K.; Wunch, D.; Deutscher, N.M.; Hase,

F.; Iraci, L.T.; Kivi, R.; Morino, I.; Notholt, J.; Ohyama, H.; Pollard, D.F.; Té, Y.; Velazco, V.A.; Warneke, T.; Sussmann, R.; Rettinger, M. Validation of Carbon Trace Gas Profile Retrievals from the NOAA-Unique Combined Atmospheric Processing System for the Cross-Track Infrared Sounder. *Remote Sens.* **2020**, *12*, 3245. https://doi.org/10.3390/rs12193245

Technical corrections:
- p. 1: TROPESS/CrIS in the title, TROPESS CrIS later in the text *Response: These are now consistently "TROPESS/CrIS".*
- p. 3, line 101: TROPOESS => TROPESS *Response: Fixed.*
- p. 5, Fig. 2: Some lines are hard to see *Response: By lines, we assume the referee means the state boundaries (solid) and the lat/lon boxes (dotted). Since these are only for reference and are not showing data, we decided to keep them as they are.*
- p. 8, Fig. 3: Axis scale is hard to read *Response: We have re-made this figure with larger font for the axes.*
- p. 9, line 323: I would suggest to add 'aircraft data' or similar: 'TROPESS CrIS CO comparisons with NOAA GML' => 'TROPESS CrIS CO comparisons with NOAA GML aircraft data' *Response: done.*
- l. 197: Calahorranol et al: 2018 => 2021 *Response: Fixed*
- l. 767 McMillam => McMillan *Response: Fixed*
- l. 876: a blank line is missing *Response: Fixed*

---

## Author Comment (AC2)

Please find our responses embedded below in *blue italics.*

Best regards,
Helen Worden & co-authors

Reviews of amt-2022-128
"TROPESS/CrIS carbon monoxide profile validation with NOAA GML and ATom in situ aircraft observations"

Anonymous Referee #2

General comments This study presents the results of validating TROPESS/CrIS carbon monoxide (CO) profiles against in-situ aircraft observations. The authors demonstrate the significance of high-resolution spaceborne CO observations and present a data quality assessment. In particular, Section 6 assists readers in how interpreting the TROPESS/CrIS data in detail. This information is essential for users applying this data to their studies. The manuscript fits the scope of the journal and is well written. I believe it can be more readable with a bit of clarification and elaboration. Therefore, I would recommend publishing this manuscript after minor revisions. Please see specific comments below.
*Response:*
*We thank the referee for their time and effort to review the paper and valuable comments that have helped to improve the manuscript.*

Specific comments
• Section 1 (Introduction): In general, more clarifications would be appreciated on the definitions of MUSES and TROPESS. Although detailed information is given in Section 2, please consider elaborating briefly on the description of the two terms (i.e., MUSES and TROPICS) in Section 1. My main questions are: Is it correct that MUSES is the algorithm name and TROPESS is the product name?
*Response:*
*That is correct, MUSES is the algorithm name and TROPESS is the new processing system and product. We agree this is a bit confusing and we have modified and added the following to Section 1, 3rd paragraph:*
The CrIS CO products evaluated here use the MUSES (MUlti-SpEctra, MUlti-SpEcies, MUlti-Sensors) algorithm (Fu et al., 2016, 2018, 2019) and are processed with the TROPESS (TRopospheric Ozone and its Precursors from Earth System Sounding), Science Data Processing System (Bowman et al., 2021). TROPESS is a NASA project that provides a framework for consistent data processing of ozone and ozone precursors across different satellite instruments.

o More specifically, on Lines 84–87: (1) Please provide full names for the abbreviations MUSES and TROPESS here. MUSES' full name is provided in Line 91, after its first appearance in Line 84. TROPESS' full name is only given in the abstract or after the conclusion section. (2) Please consider moving a part (or the entirety) of the descriptions of MUSES (Lines 99–106) and TROPESS (Lines 94–98) above Line 84.
*Response:*

*We have re-ordered the text for clarity and acronyms are now defined when first used in both the abstract and main text. We also moved some of the MUSES description in Sec. 1 to Sec. 2.1.*

• Line 238: Please provide a definition of the quality flag used here.
*Response:*
*We have added the following paragraph to Sec. 2.1 (TROPESS retrieval approach):*
The TROPESS CO products have quality flags for screening cases that did not converge or that have unphysical results. This screening checks the magnitude and spectral structure of radiance residuals, cloud retrieval characteristics, and deviation of surface emissivity from a priori values. Specifically, retrievals with good data quality of 1 have:  radiance residual standard deviation less than 12 times the radiance error, an absolute value of the radiance residual mean less than 0.7 times the radiance error, KdotDL (the normalized dot product of the Jacobians and the radiance residual) less than 0.8, LdotDL (the normalized dot product of the radiance and the residual) less than 0.6, cloud top pressures below 90 hPa, mean cloud optical depths less than 50, cloud variability (variation with respect to wavenumber) less than 3, and mean surface emissivity that did not change by more than 0.06. These threshold values are based on comparisons with in situ data and other satellite data to determine when retrievals are valid.

• Lines 329–332 and 378–380: The fact that MOPITT shows different patterns is mentioned twice in this manuscript without describing the causes. Could you provide possible reasons (e.g., differences in instruments or algorithms)?
*Response:*
*We have added the following text to Sec. 5.1:*
Since TROPESS and MOPITT retrievals both use optimal estimation algorithms and a similar prior CO error covariance, this different vertical bias pattern is most likely due to instrument differences. MOPITT uses gas filter correlation radiometry instead of spectroscopy to detect CO absorption in the atmosphere with corresponding differences in vertical sensitivity that are determined from gas cell pressure rather than spectral resolution. After accounting for retrieval differences in a priori profiles and covariances between MOPITT and IASI (another FTS instrument), George et al. (2015) find a similar positive bias for MOPITT in the upper troposphere.

*With corresponding new reference:*
George, M., Clerbaux, C., Bouarar, I., Coheur, P.-F., Deeter, M. N., Edwards, D. P., Francis, G., Gille, J. C., Hadji-Lazaro, J., Hurtmans, D., Inness, A., Mao, D., and Worden, H. M.: An examination of the long-term CO records from MOPITT and IASI: comparison of retrieval methodology, Atmos. Meas. Tech., 8, 4313–4328, https://doi.org/10.5194/amt-8-4313-2015, 2015.

• Lines 458 and 507: The phrases "what we expect for an optimal estimation" (Line 458) and "the expected behavior for optimal estimation retrievals" (Line 507) sound vague. Please elaborate on these sentences.
*Response:*
*We have made the first sentence more explicit by using Equation 1 instead of "optimal estimation" as well as a better connection to the following text that explains why it is "expected". For conclusion #7, we now have:*

Comparisons of computed observational errors and standard deviations of retrieval-aircraft comparison differences show expected vertical behavior and demonstrate significant improvement over the standard deviation of prior-aircraft differences in vertical ranges with higher retrieval sensitivity.

Technical corrections
• Line 78: CO2 to $CO_2$ *Response: Fixed.*
• Lines 85, 197, and 452: Please remove a comma before a parenthesis when presenting a reference (e.g., Hegarty et al., (2022) to Hegarty et al. (2022)). *Response: Fixed.*
• Line 147: x to x *Response: Fixed.*
• Line 188 and Table 1: How about replacing °E with °W since all longitude values are negative? *Response: Good suggestion. Done.*
• Line 433: Fig. 10 to Figure 10 *Response: Fixed.*
• Line 450: A period missing (differences As expected to differences. As expected) *Response: Fixed.*